# Fecal Microbial Diversity of Coyotes and Wild Hogs in Texas Panhandle, USA

**DOI:** 10.3390/microorganisms11051137

**Published:** 2023-04-27

**Authors:** Babafela Awosile, Chiquito Crasto, Md. Kaisar Rahman, Ian Daniel, SaraBeth Boggan, Ashley Steuer, Jason Fritzler

**Affiliations:** 1School of Veterinary Medicine, Texas Tech University, Amarillo, TX 79106, USA; 2Center for Biotechnology and Genomics, Texas Tech University, Lubbock, TX 79409, USA

**Keywords:** fecal microbiota, wildlife, 16S sequencing, coyote, hog

## Abstract

The ecology of infectious diseases involves wildlife, yet the wildlife interface is often neglected and understudied. Pathogens related to infectious diseases are often maintained within wildlife populations and can spread to livestock and humans. In this study, we explored the fecal microbiome of coyotes and wild hogs in the Texas panhandle using polymerase chain reactions and 16S sequencing methods. The fecal microbiota of coyotes was dominated by members of the phyla Bacteroidetes, Firmicutes, and Proteobacteria. At the genus taxonomic level, *Odoribacter*, *Allobaculum*, *Coprobacillus*, and *Alloprevotella* were the dominant genera of the core fecal microbiota of coyotes. While for wild hogs, the fecal microbiota was dominated by bacterial members of the phyla Bacteroidetes, Spirochaetes, Firmicutes, and Proteobacteria. Five genera, *Treponema*, *Prevotella*, *Alloprevotella*, *Vampirovibrio*, and *Sphaerochaeta,* constitute the most abundant genera of the core microbiota of wild hogs in this study. Functional profile of the microbiota of coyotes and wild hogs identified 13 and 17 human-related diseases that were statistically associated with the fecal microbiota, respectively (*p* < 0.05). Our study is a unique investigation of the microbiota using free-living wildlife in the Texas Panhandle and contributes to awareness of the role played by gastrointestinal microbiota of wild canids and hogs in infectious disease reservoir and transmission risk. This report will contribute to the lacking information on coyote and wild hog microbial communities by providing insights into their composition and ecology which may likely be different from those of captive species or domesticated animals. This study will contribute to baseline knowledge for future studies on wildlife gut microbiomes.

## 1. Introduction

The gut microbiota of wild animals contributes to host health, biology, and behavior by offering an important defense barrier against invasive pathogenic bacteria as part of a mutualistic partnership [1]. Fecal microbiomes are increasingly used as a method of wildlife health assessment for species conservation and management [2]. Fecal microbial biomarkers present a less invasive method of collecting information about wildlife populations [3]. However, despite the increased popularity, many wildlife microbiomes are underexplored [4]. In this paper, we describe the fecal microbiomes of the coyote (*Canis latrans*) and wild hogs (*Sus scrofa*), two species of ecological and economic importance in the Texas panhandle. Biodiversity in the environment and wildlife is a key factor affecting the spread of infectious diseases. The majority of zoonotic pathogens such as *Brucella* spp., *Toxoplasma gondii*, pathogenic *Escherichia coli*, and influenza virus that can be transmitted to humans are transmitted by vertebrate animals, including feral swine and coyotes [5]. Jones et al. [6] reported that the species richness of mammalian wildlife was positively correlated with the probability that pathogens would emerge from wildlife to humans. Wild hogs have been credited with billions of dollars of damages across the United States and are known to carry and transmit at least 30 bacterial, fungal, and viral diseases that threaten humans, livestock, and other wildlife species [7]. Similarly, coyotes are known to be reservoirs of numerous diseases and parasites, including rabies, distemper, and various endo and ectoparasites [8]. The panhandle region is home to 90% of the Texas cattle feedlot industry [9]. Given this high concentration of cattle production, producers work carefully to minimize the spread of disease between wildlife and their own animals [10]. The Texas panhandle is made up of a variety of landscapes, most notably: expansive plains, and rugged canyon lands. These landscapes provide many resources for a variety of wildlife populations. Domestic and wild animal land use overlap is almost a guarantee [10]. This study aims to describe and explore the fecal microbiome in coyotes and wild hogs in the Texas panhandle.

## 2. Methodology

### 2.1. Fecal Sample Collection

This descriptive study was conducted from March to May 2021. Fecal samples of coyotes (*n* = 19) and wild hogs (*n* = 19) were acquired opportunistically from the USDA-Wildlife Services, Texas A&M Agrilife. Approximately 50 g of feces were collected from each carcass and placed in an airtight sterile sample container and transported to the lab. The samples were then placed in 50 mL sterile centrifuge tubes and stored at 4 °C until DNA extraction.

### 2.2. DNA Extraction, Amplification, and Sequencing

Fecal genomic DNA was extracted for each sample using the Qiagen QIAamp Fast DNA stool Mini kits following the manufacturer’s instructions (Thermo Fischer Scientific, Waltham, MA, USA). DNA samples were further quantified using a NanoDrop ND-1000 spectrophotometer (Thermo Fisher Scientific, MA, USA). The DNA samples were stored at −20 °C until further processing. The 16S library preparation and sequencing were conducted at the Texas Tech University Genomic Center, Lubbock, TX, USA. The V3 and V4 hypervariable regions of the bacterial 16S rRNA gene were amplified by polymerase chain reaction (PCR). The protocol includes the primer pair sequences for the V3 and V4 regions that create a single amplicon of approximately ~550 bp as previously described by Klindworth et al. [11]. PCR were performed using 2.5 μL (5 ng/μL in 10 mM Tris pH 8.5) fecal microbial genomic DNA per sample, 5 μL amplicon PCR forward and reverse primers each at 1 μM/L, 12.5 μL repliQua HiFi ToughMix (QuantaBio, Beverly, MA, USA) per sample to a total volume of 25 μL per sample. PCR protocols were as follows: initial denaturation at 95 °C for 3 min, followed by 25 cycles of 95 °C for 30 s, 55 °C for 30 s, and 72 °C for 30 s, followed by a final extension at 72 °C for 5 min and then held at 4 °C. PCR products were quantified on a 1 μL of each PCR product was run on a Bioanalyzer DNA 1000 chip to verify the size. Using the V3 and V4 primer pairs in the protocol, the expected size on a Bioanalyzer trace after the Amplicon PCR step is ~550 bp. Cluster generation and DNA sequencing were performed using Illumina 600V3 chemistry on a MiSeq platform.

### 2.3. Bioinformatics Analysis

A total of 19 samples were processed, each for coyote and hog. Initial bioinformatic analysis was performed by the sequencing lab of the Texas Tech University Genomic Center, Lubbock, TX, United States. To identify the bacteria involved, the sequenced reads were then mapped and aligned against the 16S genes of approximately 3.8 million individual bacteria downloaded and archived from the Ribosomal Database Project (RDP) housed at the Center for Microbial Ecology at Michigan State University (http://rdp.cme.msu.edu/, accessed on 23 April 2022). The reads were mapped and aligned using the metagenomics and RNA-Seq components of the DNA-Star Lasergene genomics analysis software (https://www.dnastar.com/, accessed on 23 April 2022). A customized PERL (Practical Extraction and Report Language) program was developed to filter out bacteria whose 16S genes aligned for samples whose raw read counts were too few to be considered statistically significant. This program removed bacteria for which all the samples had fewer than 10 reads mapped and aligned to the 16S gene sequences for these bacteria. Another customized PERL program processed the read information for the samples to identify the Operational Taxonomic Unit (OTU). The OTU for every sample was resolved to the genus taxonomic level; however, not in all cases. The information retrieved includes the RDP identifier, the read counts for each sample, and the taxonomic breakdown up to the OTU level. Raw sequence data for this project is available in the National Center for Biotechnology Information (NCBI) Sequence Read Archive (SRA), Bioproject PRJNA921837.

### 2.4. Data Analysis

Data management was carried out using Microsoft Excel 365 for Windows (2021, Microsoft Corp., Redmond, WA, USA) and then imported and analyzed using R software using different R packages (v4.1.1). Data analysis was conducted as previously described [12], separately for coyotes and wild hogs without comparison. Initial descriptive statistics were performed to explore the distributions of OTU reads and the taxonomic ranks. Further analysis was conducted to determine the relative abundances for the total reads for the identified OTUs for coyotes and wild hogs using bar plots and boxplots. We assessed Alpha diversity using Shannon’s Diversity Index as a measure of microbial richness and evenness in each sample, and Chao’s Richness estimate as a measure of the uniqueness or richness of species in each sample. Beta-diversity was evaluated using the Bray–Curtis dissimilarity index and agglomerative hierarchical clustering using Ward’s method. Beta diversity was visualized using a distance matrix heatmap and hierarchical clustering dendrogram. Heat trees, describing the core microbial community for coyotes and wild hogs, were generated separately from a neighbor-joining tree of OTUs and associated relative abundances across the taxonomic ranks. We explored the predictive functional profile of the microbiota of coyotes and wild hogs using MicrobiomeAnalyst. MicrobiomeAnalyst is a web-based platform for comprehensive statistical, visual, and meta-analysis of microbiome data as previously described [13].

## 3. Result

A total of 141,206 and 78,723 read sequences were generated following the Next Generation Sequencing for the expression of the bacterial 16S gene for coyotes and wild hogs, respectively. After the filtering and cleanup step, 134,914 sequence reads from coyote samples were mapped to the 16S gene database resulting in 506 unique OTUs representing 20 phyla, 46 classes, 68 orders, 60 families, and 91 genera. Median sequence reads (and associated inter-quantile range) for coyotes was 4233 ± 8805 per sample from the 506 unique OTUs. A total of 769,641 sequence reads from wild hog samples were mapped to the 16S gene database resulting in 807 unique OTUs representing 24 phyla, 51 classes, 71 orders, 65 families, and 106 genera. Median sequence reads (and associated inter-quantile range) for wild hogs was 42,320 ± 28,531.5 per sample from the 807 unique OTUs. The fecal microbiota of coyotes was dominated by members of the phyla *Bacteroidetes* (Median% ± IQR: 30.81 ± 28.71), *Firmicutes* (Median% ± IQR: 23.33 ± 19.36), and *Proteobacteria* (Median% ± IQR: 3.97 ± 13.73) across the samples (Figure 1 and Appendix A). At the genus taxonomic level, *Odoribacter* (Median% ± IQR: 17.92 ± 39.64), *Allobaculum* (Median% ± IQR: 11.82 ± 14.40), *Coprobacillus* (Median% ± IQR: 4.61 ± 7.77), and *Alloprevotella* (Median% ± IQR: 2.15 ± 5.06) were the dominant genera in fecal microbiome of coyotes in the Texas panhandle (Appendix A). These four dominant genera also constitute the most abundant genera of the core microbiota of coyotes (Figure 2). The alpha diversity indices are presented in Appendix A. Chao’s estimated species richness values among the coyote samples vary from sample to sample with an average richness of 12 ± 2.75 SD per sample. The Shannon index estimate, as a measure of both richness and evenness, also varies from sample to sample consistent with the Chao index. The median Shannon estimate per sample (±IQR) was 1.14 ± 0.82. The Bray–Curtis dissimilarity index and the hierarchical clustering (Appendix A) showed the samples clustered into two clades based on how dissimilar and non-correlated the coyote samples were concerning microbial richness and abundance. Using the MicrobiomeAnalyst platform for the predictive functional profile of the coyote microbiota, a total of 76 human-related diseases were associated with the coyote microbiota reported in this study, either contributing to the increasing or decreasing risk of diseases in humans (Appendix A). However, after adjusting for the false discovery rate, 13 human-related diseases including Hepatitis B, type-1 diabetes, cancer, obesity, and schistosomiasis were statistically associated with the coyote microbiota (Appendix A).

The fecal microbiota of wild hogs was dominated by bacterial members of the phyla *Bacteroidetes* (Median% ± IQR: 54.38 ± 20.84), *Spirochaetes* (Median% ± IQR: 14.31 ± 17.54), *Firmicutes* (Median% ± IQR: 9.93 ± 4.88) and *Proteobacteria* (Median% ± IQR: 5.45 ± 6.35) across the samples (Figure 3 and Appendix A). At the genus taxonomic level, *Treponema* (Median% ± IQR: 19.71 ± 31.30), *Prevotella* (Median% ± IQR: 15.85 ± 17.18), *Alloprevotella* (Median% ± IQR: 8.32 ± 17.19), *Vampirovibrio* (Median% ± IQR: 7.52 ± 7.59) and *Sphaerochaeta* (Median% ± IQR: 5.86 ± 7.64) were the dominant genera in fecal microbiome of wild hogs in the Texas panhandle (Appendix A). These five dominant genera also constitute the most abundant genera of the core microbiota of wild hogs in this study (Figure 4). The alpha diversity indices for wild hog samples are presented in Appendix A. Chao’s estimated species richness values among the wild hog samples vary from sample to sample with an average richness of 14.97 ± 3.21 SD per sample. The Shannon index estimate, as a measure of both richness and evenness, also varies from sample to sample consistent, with the Chao index. The median Shannon estimate per sample (±IQR) was 1.21 ± 0.38. The Bray–Curtis dissimilarity index and the hierarchical clustering dendrogram (Appendix A) showed the samples clustered into three clades based on how dissimilar and non-correlated the coyote samples were concerning microbial richness and abundance. Using the MicrobiomeAnalyst platform for the predictive functional profile of the wild microbiota, a total of 68 human-related diseases were associated with wild hog microbiota reported in this study, either contributing to the increasing or decreasing risk of diseases in humans (Appendix A). However, after adjusting for the false discovery rate, 17 human-related diseases, including Crohn’s disease, cancer, type-1-diabetes, and ulcerative colitis, were statistically associated with the wild hog microbiota (Appendix A).

## 4. Discussion

In this study, we described the fecal microbial ecology of important wildlife species that are often seen at one health interface. Both wildlife species are often part of wildlife control program in the United States, in part due to their environmental nuisance potential, agricultural and domestic damages, and possible transmission of infectious diseases at one health interface. While wildlife can serve as reservoirs of infectious diseases, information on other important culturally independent and culturally dependent bacteria with the potential to be beneficial and challenges to the wildlife, domestic animals, and humans can be elucidated through fecal microbial ecological studies. In addition, the ecology of infectious diseases requires the involvement of wildlife, yet the wildlife interface is often neglected and understudied. Pathogens related to infectious diseases are often maintained within wildlife populations and can spread to livestock, which can lead to an outbreak of the disease in animals and humans [6,14] Furthermore, descriptive study on fecal microbial ecology of wildlife is very important in further strengthening the importance of gut microbiota in wildlife conservation and adaptation, especially in relation to the protection against infectious disease occurrence among wildlife species. We do not usually observe wildlife species with diabetes, obesity, diarrhea, or other gastrointestinal diseases, therefore, elucidating the nature of the wildlife microbiota may provide useful information to human and animal health on what types of microbiotas are contributing to that protection from gastroenteric diseases and other metabolic diseases. Such knowledge can be applied to human and animal health, in addition to wildlife conservations. Additionally, such knowledge of microbiotas as provided in this study, may shed more light on how some environmental and socioeconomic factors, such as wildlife hunting, destruction of the forest for agriculture and habitation, urbanization of wildlife and wildlife environment, change in the diet of wildlife due to human feeding of wildlife (e.g., coyotes), and changes in agricultural or food production practices, influences the gut microbiotas of wildlife species [15].

In this study, we characterized the microbial diversity and taxonomic composition of fecal samples of coyotes (*Canis latrans*) and wild hogs (*Sus scrofa*). We detected unique Operational Taxonomic Units (OTUs) from coyotes and wild hogs which included several phyla, classes, orders, families, and genera. Both coyotes and wild hogs shared most of the same taxonomic groups of gut microbiomes which are common in most vertebrate animals [16]. We detected major fecal microbiota phyla (Bacteroidetes, Firmicutes, and Proteobacteria), which are commonly associated with the gut microbiotas of wolves and dogs [17,18]. Among the major microbiota, we detected Bacteroidetes as the most dominant phyla; however, in previous studies, Firmicutes and Fusobacteria were identified as the major phyla of the canine gut microbiome [17,18,19]. Both Firmicutes and Bacteroidetes are widely accepted and considered to be significant in the maintenance of intestinal homeostasis and an important biomarker for gut dysbiosis [17,18]. Among the *Firmicutes*, several genera constitute canine microbiotas, including the obligate anaerobic, spore-forming members of *Clostridium* species. *Clostridium* species are ubiquitous, asymptomatic, and pathogenic bacteria of medical and veterinary importance with the potential to manifest life-threatening gastrointestinal conditions. Strains of *Clostridium difficile*, a member of the *Firmicutes*, has been reported as a leading cause of antibiotic and nosocomial- associated diarrhea infections in humans [20,21].

*Odoribacter* was found to be the most abundant bacterial genus in the fecal samples of coyotes. These results agreed with a previous study where the *Odoribacter* genus was reported as an oral organism in canids [22]. *Odoribacter* organisms are pathogenic in humans and domestic animals and can cause abdominal abscesses and periodontitis in domestic animals [23]. Additionally, the bacteria of the genus *Odoribacter* have previously been identified as opportunistic pathogens, causing dysbiosis and microbiome-linked infections such as ulcerative colitis [24]. Another frequently occurring bacterial genus in coyotes was *Allobaculum*. *Allobaculum* has been previously identified in canine feces, and the bacterium was reported to produce a large repertoire of mucin O-glycan targeting enzymes that can degrade the mucus layer in the gut of animals and humans [25]. Mucosal colonization by *Allobaculum mucolyticum*, a member of genus *Allobaculum* have been connected with inflammatory bowel disease and are potentially immunogenic—playing critical roles in intestinal inflammation [25]. Other bacteria identified in coyote microbiota have also been associated with some clinical conditions in humans. For instance, the genus *Alloprevotella* was reported to be associated with Rheumatoid arthritis (RA) disease in microbiome studies of early RA patients [26]. In addition, *Coprobacillus catenaformis*, the only known member of the genus *Coprobacillus* is an opportunistic pathogen in gastrointestinal disease and a causative agent of bacteremia in humans and mice [27]. Additionally, *Bacteroides* have been reported as an important genus with both beneficial and detrimental properties, especially anaerobic pathologies outside the host [28].

Specifically in wild hogs, Bacteroidetes, Spirochaetes, Firmicutes, and Proteobacteria were the most common phyla detected, which agreed with the previous result of Kim et al. [29] and [30]. The dominant bacterial genus in the wild hog microbiome, in order of decreasing abundance, were *Treponema*, *Prevotella*, *Alloprevotella*, *Vampirovibrio*, and *Sphaerochaeta*. In this study, *Treponema* and *Prevotella* were detected as the leading genera for wild hogs’ fecal microbiomes. Similarly, *Prevotella* is the representative organism in the stomach, cecum, colon, and rectum in pigs [31]. On the other hand, *Prevotella* have been associated with localized mucosal and systemic diseases in humans, such as acute necrotizing ulcerative gingivitis, adult periodontitis, metabolic disorders, low-grade systemic inflammation, and cellulitis [32,33]. Higher numbers of *Prevotella* have been associated with chronic gut inflammatory conditions in human immunodeficiency virus patients [34]. Similarly, species of *Alloprevotella* and *Prevotella* genus have been isolated in the oral microbiomes of human and animal patients [35]. Oral anaerobic *Prevotella* serves as commensals in the oral environment; however, their colonization on extra-oral sites is associated with harmful roles in gut and respiratory health and infections [36]. Excessive growth of *Prevotella* in the female genital mucosal surface show an association with bacterial vaginosis [37]. *Treponema hyodysenteriae*, *Treponema berlinense*, *Treponema succinifaciens*, and *Treponema porcinum* were previously isolated from the gut of swine [38,39]. Treponemal infection may cause the clinical manifestation of various diseases such as venereal syphilis, endemic syphilis, yaws, and pinta in humans [40]. We found the average microbial species richness and evenness were lower in coyote samples compared to wild hog samples. This agrees with and is supported by the previous findings of Ley et al. [41], who expressed that omnivorous and herbivorous have higher genus-level richness and bacterial diversity than that carnivorous. This bacterial diversity maybe influenced by the host diet and phylogeny [42].

We also explored the functional profiles of the microbiota of both wild hogs and coyotes using the publicly available web-based MicrobiomeAnalyst in an attempt to gain more knowledge into the important functionalities of the gut microbiota of the wildlife species observed in this study. Some of the microbiotas of the wild hogs and coyotes have been associated with certain human-related diseases. We found an increased risk of susceptibility to Malaria, type I diabetics, Resistance to PD-L1 monoclonal antibodies, depression, Urogenital schistosomiasis, Colorectal Cancer, and resistance to an immune checkpoint inhibitor consistent with previous reports [43,44,45]. Concurrently, human and wildlife diseases have been identified to be connected to the gut microbiota where alterations in the gut microbes can result in gastrointestinal health outcomes, such as Crohn’s disease [46], and inflammatory bowel disease [47], consistent with the functional profile of microbiome of both wild hogs and coyotes reported in this study. This finding further shows the importance of these microbiota in contributing to the increasing or decreasing risk of important infectious, non-infectious, and metabolic diseases in domestic animals and humans. In addition, this functional characteristic may be important in comparative microbiome of human and wildlife species, such as coyotes that frequent human environments, and may help in elucidating the role of human feeding of coyotes in shaping the gut microbiota of wildlife. On the other hand, it is important to note that the fecal microbiotas of coyotes and wild hogs may consist of known culturally dependent zoonotic bacteria, such as Salmonella, Brucella, Yersinia, etc., and potentially unknown and novel culturally independent bacteria that may be of zoonotic significance. Therefore, environments shared by humans, animals and wildlife can be contaminated by the feces containing these microbiotas. While we used the publicly available web-based MicrobiomeAnalyst to gain more knowledge into the functionalities of the microbiota, it is important to emphasize the limitation of such a tool in terms of applicability and reliability. Finding an association of human diseases with sequences affiliated to a microbiota genus, particularly genera such as Prevotella, Lactobacillus or Bacteroides, are not specific enough. Each genus consists of many different species that may be related based on DNA sequences, however, with different metabolic functions. Secondly, it seems very unlikely that diseases such as diabetes or colorectal cancer are affected by the genera listed–it is more likely that differences in these bacterial groups is a secondary effect rather than a cause of the disease. Furthermore, these potential associations were identified in patients directly affected by the diseases, not through indirect associations as conducted in this study. Therefore, such associations in this study are likely to be spurious, and inferences made in this case are speculative. More studies are needed to elucidate the functional characteristics of wildlife microbiota for a better understanding of their roles in preventing or contributing to some wildlife diseases.

## 5. Conclusions

In conclusion, we described the fecal microbiota of free-roaming coyotes and wild hogs in West Texas. The fecal microbiota provides information on diverse and abundant microbial communities that were unculturable and indescribable by culture-dependent methods. We detected dominant phyla (Bacteroidetes, Firmicutes, and Proteobacteria) across the microbiome of coyotes and wild hogs. The four dominant genera identified (*Odoribacter, Allobaculum, Coprobacillus,* and *Alloprevotella*) also constitute the most abundant genera of the core microbiota of coyotes. The information provided in this study may be important in wildlife conservation, wildlife adaptation, fecal microbiota transplantation, and gut dysbiosis due to wildlife diseases and may also serve as baseline information for future studies, as well as comparative microbiota studies with domestic canids and swine. This report will contribute to the lacking information on coyote and wild hog microbial communities by providing insights into their composition, structure, and ecology which may likely be different from those of captive species or domesticated animals.

## Figures and Tables

**Figure 1 microorganisms-11-01137-f001:**
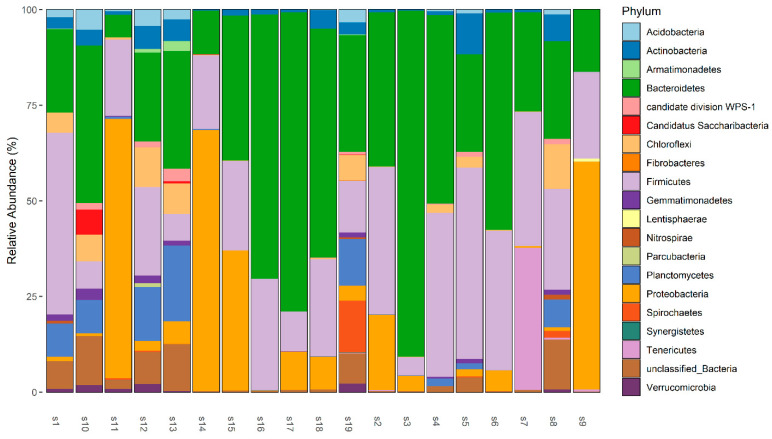
Phyla level relative abundances of fecal microbiome in 19 coyote samples.

**Figure 2 microorganisms-11-01137-f002:**
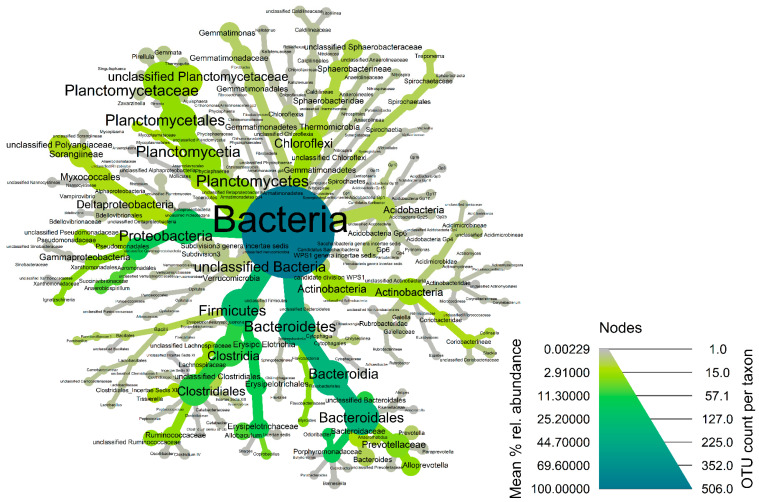
Heat-tree of the core fecal microbiota of coyotes in the Texas panhandle. Node and line widths increase with increasing OTU counts while increasing green color indicates increasing mean % relative abundance of the core taxonomic groups.

**Figure 3 microorganisms-11-01137-f003:**
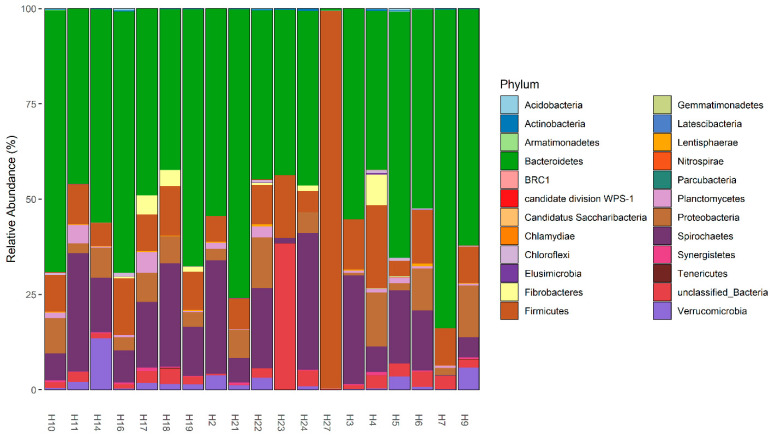
Phyla level relative abundances of fecal microbiome in 19 wild hog samples.

**Figure 4 microorganisms-11-01137-f004:**
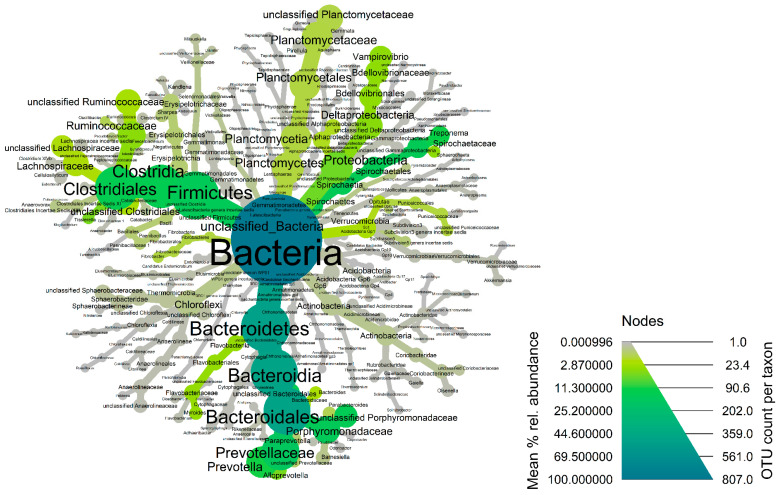
Heat-tree of the core fecal microbiota of wild hogs in the Texas panhandle. Node and line widths increase with increasing OTU counts while increasing green color indicates increasing mean % relative abundance of the core taxonomic groups.

## Data Availability

Raw sequence data for this project is available in the National Center for Biotechnology Information (NCBI) Sequence Read Archive (SRA), Bioproject PRJNA921837.

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
