# Peer review of "Fecal Microbial Diversity of Coyotes and Wild Hogs in Texas Panhandle, USA"

_microorganisms, 2023, doi:10.3390/microorganisms11051137_

Round 1
Reviewer 1 Report
In my opinion, the article titled ‘Fecal microbial diversity of coyotes and wild hogs in Texas panhandle’ is complete I have only a few minor comments and congrats to the Authors.
Minor comments:
Line 40 and 46: ‘Wild pigs’ In all text the Authors use ‘wild hogs’ maybe the ‘wild pigs’ should be changed to ‘wild hogs’
Line 78 and 81: ‘PCR reaction’ remove ‘reaction’
Line 90: ’16 S’ remove space
Line 305: ‘inflammatory’
Author Response
Thanks for your review.
In my opinion, the article titled ‘Fecal microbial diversity of coyotes and wild hogs in Texas panhandle’ is complete I have only a few minor comments and congrats to the Authors.
Minor comments:
Line 40 and 46: ‘Wild pigs’ In all text the Authors use ‘wild hogs’ maybe the ‘wild pigs’ should be changed to ‘wild hogs’
AU: Thanks we corrected this through out the manuscript.
Line 78 and 81: ‘PCR reaction’ remove ‘reaction’
AU: We deleted reaction
Line 90: ’16 S’ remove space
AU: We removed the space
Line 305: ‘inflammatory’
AU: Done
Reviewer 2 Report
Dear author(s)
Thanks for a unique research paper and presentation.
However, there are some inquiries that need your answer.
1. The possibility of the direct between these wild animals (coyotes or hogs) and human in the wild environment is rare. So, how their pathogenic microbiota can be transmitted to human causing direct infection? Is it indirect infection?
2. The title “Fecal microbial diversity of Coyotes and Wild Hogs in Texas panhandle”. The country could be added to the title “Fecal microbial diversity of Coyotes and Wild Hogs in Texas panhandle, USA”.
3. Line 236, (Middelbos et al., 2010; Suchodolski et al., 2008; Zhang & Chen, 2010). If you arrange the reference chronologically, please, re-arrange them (Suchodolski et al., 2008; Middelbos et al., 2010; Zhang & Chen, 2010). The same in line 302 (Kim & Lee, 2021; Zheng et al., 2018; Zhou et al., 2022).
4. Line 252, ulcerative colitis (UC). If this abbreviation was not mentioned again the manuscript, it should be deleted. The same, inflammatory bowel disease (IBD) in line 306.
4. The nomenclature of the bacteria should be written in a correct and the same way all over the manuscript.
5. Line 273, “Similarly, Prevotella asthe representative organism”. The sentence should be corrected as “Similarly, Prevotella is the representative organism”.
6. Line 277, (Aguilera et al., 1998; Cantas & Suer). Please, complete the year of publishing (Cantas & Suer).
7. Some abbreviation should be mentioned as full words at the first mention such as (HVT in line 279).
With my best wishes
Author Response
Thank you for your review and observations. We appreciate it.
- The possibility of the direct between these wild animals (coyotes or hogs) and human in the wild environment is rare. So, how their pathogenic microbiota can be transmitted to human causing direct infection? Is it indirect infection?
AU: Both direct and indirect exposure is possible considering people hunt these wildlife species and these wildlife species frequently stray into the human environment especially the livestock farms in Texas. This is one of the reasons, the wild swine control program has been established in the USA. So, I will say both direct and indirect exposure is possible. Thanks
- The title “Fecal microbial diversity of Coyotes and Wild Hogs in Texas panhandle”. The country could be added to the title “Fecal microbial diversity of Coyotes and Wild Hogs in Texas panhandle, USA”.
AU: Thanks, we have added USA to the title
- Line 236, (Middelbos et al., 2010; Suchodolski et al., 2008; Zhang & Chen, 2010). If you arrange the reference chronologically, please, re-arrange them (Suchodolski et al., 2008; Middelbos et al., 2010; Zhang & Chen, 2010). The same in line 302 (Kim & Lee, 2021; Zheng et al., 2018; Zhou et al., 2022).
AU: Thank you, we have rearranged the references.
- Line 252, ulcerative colitis (UC). If this abbreviation was not mentioned again the manuscript, it should be deleted. The same, inflammatory bowel disease (IBD) in line 306.
AU: Thanks, we deleted the abbreviations
- The nomenclature of the bacteria should be written in a correct and the same way all over the manuscript.
AU: Thanks, we have checked through the manuscript for consistency of the bacteria names.
- Line 273, “Similarly, Prevotella asthe representative organism”. The sentence should be corrected as “Similarly, Prevotella is the representative organism”.
AU: THanks, we corrected the errors
- Line 277, (Aguilera et al., 1998; Cantas & Suer). Please, complete the year of publishing (Cantas & Suer).
AU: We corrected this as Cantas & Suer, 2014
- Some abbreviation should be mentioned as full words at the first mention such as (HVT in line 279).
AU: Thank you we wrote HIV in full
With my best wishes